



**Ensemble streamflow forecasting over a cascade reservoir catchment with**
**integrated hydrometeorological modeling and machine learning**
Junjiang Liu[1], Xing Yuan[1*], Junhan Zeng[1], Yang Jiao[1], Yong Li[2], Lihua Zhong[2], Ling
Yao[3]
[1]School of Hydrology and Water Resources, Nanjing University of Information
Science and Technology, Nanjing 210044, China
[2]Guangxi Meteorological Disaster Prevention Center, Nanning 530022, China
[3]Guangxi Guiguan Electric Power Co., Ltd., Nanning 530029, China

*Corresponding author address:* Xing Yuan, School of Hydrology and Water Resources, Nanjing University of Information Science and Technology, Nanjing 210044, China E-mail: xyuan@nuist.edu.cn



**Abstract.** A popular way to forecast streamflow is to use bias-corrected
meteorological forecast to drive a calibrated hydrological model, but these
hydrometeorological approaches have deficiency over small catchments due to
uncertainty in meteorological forecasts and errors from hydrological models,
especially over catchments that are regulated by dams and reservoirs. For a cascade
reservoir catchment, the discharge of the upstream reservoir contributes to an
important part of the streamflow over the downstream areas, which makes it
tremendously hard to explore the added value of meteorological forecasts. Here, we
integrate the meteorological forecast, land surface hydrological model simulation and
machine learning to forecast hourly streamflow over the Yantan catchment, where the
streamflow is influenced both by the upstream reservoir water release and the
rainfall-runoff processes within the catchment. Evaluation of the hourly streamflow
hindcasts during the rainy seasons of 2013-2017 shows that the hydrometeorological
ensemble forecast approach reduces probabilistic forecast error by 10% and
deterministic forecast error by 6% as compared with the traditional ensemble
streamflow prediction (ESP) approach during the first 7 days. The deterministic
forecast error can be further reduced by 6% in the first 72 hours when combining the
hydrometeorological forecast with the long short-term memory (LSTM) deep learning
method. However, the forecast skill for LSTM using only historical observations
drops sharply after the first 24 hours. This study implies the potential of improving
flood forecast over a cascade reservoir catchment by integrating meteorological
forecast, hydrological modeling and machine learning.





**Keywords**: Streamflow; Hydrological modeling; LSTM; Reservoir; Ensemble
forecast



## 1. Introduction

Flood events are the most destructive ones among the natural disasters, causing huge damages to human society. Reservoirs are massively constructed to regulate river flows, which has significantly reduced flood risks or damages (Ji et al., 2020). However, the number and intensity of precipitation extreme events are increasing in many areas as the global warming continues, thus amplify the potential of flood hazards (Hao et al., 2013; Shao et al., 2016; Wei et al., 2018; Yuan et al., 2018a; Wang et al., 2019). Accurate streamflow forecast is thus needed to provide guidelines for reservoir operations (Robertson et al., 2013), especially when the flood risk is increasing under global warming.

A common approach of streamflow forecast is to use hydrological models, where the first attempt could be traced back to 1850s, using simple regression-type approaches to predict discharge from observed precipitation (Mulvaney, 1850). Since then, model concepts have been further augmented by designing new data networks, addressing heterogeneity of hydrological processes, capturing the nonlinear characteristics of hydrologic system and parameterizing models (Hornberger and Boyer, 1995; Kirchner, 2006). With the advancements of computer technology and high-resolution observation, a well-parameterized hydrological model can simulate streamflow with high accuracy (Kollet et al., 2010; Ye et al., 2014; Graaf et al., 2015; Yuan et al., 2018b).



Streamflow simulations from hydrological models heavily rely on
meteorological forcing inputs, especially precipitation, which can be measured at
in-situ gauges or retrieved from satellites and radars. However, for medium-range (2–
15 days ahead) streamflow forecasts, precipitation forecast is needed (Hopson et al.,
2002). To improve the forecast, ensemble techniques that can give a deterministic
estimate as well as its uncertainty became popular. Ensemble weather forecasting can
be traced back to 1963 when Leith transferred a deterministic forecast into an
ensemble using the Monte-Carlo method to describe the atmospheric uncertainty
(Leith, 1963). In the 1990s, ensemble forecasting was developed into an integral part
of numerical weather prediction, which showed higher skill than the deterministic
forecast even with higher model resolution (Toth et al., 2001). Due to its rapid
development, ensemble weather forecasts and climate predictions are applied to
hydrological forecasting studies by combining with hydrological models (Jasper et al.,
2002; Balint et al., 2006; Jaun et al., 2008; Xu et al., 2015; Yuan et al., 2016; Zhu et
al., 2019). Provided with streamflow variability, a reservoir can maintain a reliable
utility from natural streamflow better than provided with a deterministic streamflow
forecast (Zhao et al., 2011). However, the streamflow prediction skill depends on
whether the precipitation forecasts introduced into the hydrological model are skillful
(Alfieri et al., 2013). When assessing the skill of this hydrometeorological forecast
approach, a benchmark is needed. Using ensembles of historical climatology data
(Day, 1985) as meteorological forecast inputs, which is known as ensemble
streamflow prediction (ESP), is often selected as the benchmark approach.



Evaluations of hydrological forecasts indicated that forecast skill has a close
relationship with catchment size, geographical locations and resolutions (Alfieri et al.,
2013; Pappenberger et al., 2015), which means there is a necessity to compare with
the ESP to show the skill of the hydrometeorological forecast approach.

88        Although physically based hydrological models are widely used, it is still hard

to apply a hyper-resolution distributed model for streamflow forecasting due to its
demand for observation data, complex model structures and computational resources
requirements for calibration and application (Wood et al., 2011; Kratzert et al., 2018;
Yaseen et al., 2018). In cascade reservoir systems, there are two sources of streamflow,
one is from the rainfall within the interval basin and the other is from the upstream
reservoir discharge. While the rainfall-runoff relationship is well studied, it is
challenging to reproduce the reservoir operating rules in a physical model (Gao et al.,
2010; Zhang et al., 2016; Dang et al., 2020).

97        Machine learning methods can recognize patterns hidden in input data and can

simulate or predict streamflow without explicit descriptions of the underlying physical
processes (Kisi et al., 2007; Adnan et al., 2019). Neural networks are suitable for
streamflow forecasting among machine learning models, some of them can even
outperform physically based hydrological models. For example, Humphrey et al.
(2016) showed that their combined Bayesian artificial neural network with GR4J
model approach outperforms the GR4J model in monthly streamflow forecasting.
Kratzert et al. (2019) showed that the long short-term memory (LSTM)-based



approach outperforms a well-calibrated Sacramento Soil Moisture Accounting Model
(SAC-SMA). Yang et al. (2020) used the geomorphology-based hydrological model
(GBHM) combined with traditional ANN model to simulate daily streamflow, which
can provide enough physical evidence and can run with less observation data.
Although neural network models are criticized with little physical evidence (Abrahart
et al., 2012), their potential in hydrological forecasting is yet to be explored.
In this study, we combine the machine learning with hydrometeorological
approach for hourly streamflow forecast over a data-limited cascade reservoir
catchment located in southwestern China. We use the TIGGE-ECMWF
meteorological forecasts to drive a newly developed CSSPv2 high-resolution land
surface model (Yuan et al., 2018) to provide runoff and streamflow forecasts, and
adjust the results via LSTM model to improve streamflow forecast. We strive to (1)
calibrate the hydrological model, (2) bias correct the meteorological forecasts, (3)
evaluate the streamflow forecast skill and (4) test the physical-statistical combined
approach.
**2. Study Area, Data, Model and Method**
*2.1 Study Area*
The Yantan Hydropower Station is in the middle reaches of Hongshui River in
Dahua Yao Autonomous County, Guangxi Province. The Yantan Hydropower Station
is the fifth level in the 10-level development of Hongshuihe hydropower base in
Nanpanjiang River, connected with upstream Longtan Hydropower Station and the



downstream Dahua Hydropower Station. The drainage area between the Longtan
Hydropower Station and Yantan Hydropower Station is 8,900 km$^2$. The annual mean
streamflow at Yantan gauge is 55.5 billion m$^3$. The river passes through karst
mountain area, with narrow valley, steep slope and scattered cultivated land, and the
average slope is 0.036%. Figure 1 shows the locations of 4 hydrological gauges, with
detailed information listed in Table 1.
*2.2 Data and Method*
*2.2.1 Hydrometeorological observations*
There are 97 meteorological observation stations within the catchment (Figure
1). Here, observed hourly 2m-temperature, 10m-wind speed, relative humidity,
accumulated precipitation and surface pressure data were interpolated into a 5km
gridded observation dataset via inverse distance weight method. The hourly surface
downward solar radiation data from China Meteorological Administration Land Data
Assimilation System (CLDAS) was also interpolated into 5km via bilinear
interpolation method. The hourly surface downward thermal radiation (long) was
estimated by specific humidity, pressure, temperature. This dataset was used to drive
the CSSPv2 land surface hydrological model.
The monthly runoff for each 5km grid was estimated by disaggregating control
streamflow station observations with the ratio of observed grid monthly precipitation
and catchment mean precipitation. The gridded runoff was used to calibrate the
CSSPv2 model at each grid (Yuan et al., 2016).





*2.2.2 Ensemble Meteorological hindcast data and ESP hindcasts*
The TIGGE dataset consists of ensemble forecast data from 10 global Numerical
Weather Prediction centers started from October 2006, which has been made available
for scientific research, via data archive portals at ECMWF and CMA. TIGGE has
become a focal point for a range of research projects, including research on ensemble
forecasting, predictability, and the development of products to improve the prediction
of severe weather (Bougeault et al., 2010). In this paper, TIGGE data from April to
September during 2013-2017 from ECMWF were used as meteorological hindcast
data. The 3-hourly meteorological hindcasts for 7-day lead time from 51 ensemble
members (including control forecast) were interpolated into 5km resolution via
bilinear interpolation. The forecast precipitation and temperature were corrected to
match the observational means to remove the biases.
The ESP was accomplished by applying historical meteorological forcings (Day,
1985). In this paper, the meteorological forcings from the same date as the forecast
start date to the next 9 days of each year (excluding the target year) were selected as
the ESP forcings. Take April $1^{st}$, 2013 as example, the 7-day observations started from
April $1^{st}$ to April $10^{th}$ (i.e., April $1^{st}$-April $7^{th}$, April $2^{nd}$-April $8^{th}$, …, April $10^{th}$-April
$16^{th}$) in the year of 2014, 2015, 2016 and 2017 were selected as the forecast ensemble
forcings of the issue date (April $1^{st}$), with a total of 40 ensemble members.
*2.2.3 CSSPv2 streamflow hindcasts*



The physical hydrological model used in this paper is the Conjunctive
Surface-Subsurface Process model version 2 (CSSPv2; Yuan et al., 2018). The
CSSPv2 model is a distributed, grid-based land surface hydrological model, which
was developed from the Common Land Model (Dai et al., 2003, 2004), but with better
representations in lateral surface and subsurface hydrological processes and their
interactions. The routing model used here employs the kinetic wave equation as
covariance function, which is solved via a Newton algorithm. A main reason for
adopting this covariance function is that it suits the basin with mountainous terrain.
The CSSPv2 model was successfully used to perform a high-resolution (3 km) land
surface simulation over the Sanjiangyuan region, which is the headwater of major
Chinese rivers (Ji and Yuan, 2018). In this paper, we calibrated CSSPv2 model against
monthly estimated runoff to simulate the natural hydrological processes by using the
SCE-UA approach (Duan et al., 1994). The calibrated parameters include maximum
velocity of baseflow, variable infiltration curve parameter, fraction of maximum soil
moisture where non-linear baseflow occurs and fraction of maximum velocity of
baseflow where non-linear baseflow begins. The hourly observed streamflow at
Yantan hydrological gauge was used to calibrate the CSSPv2 routing model manually,
including slope, river density, roughness, width and depth. The observed streamflow
at Longtan hydrological gauge were added into the corresponding grid to provide
upstream streamflow information. The simulation results were evaluated by
calculating the Nash-Sutcliffe efficiency (NSE) with corresponding observation data.





After calibration, we drove the CSSPv2 model using 5km regridded and
bias-corrected TIGGE-ECMWF forecast forcing during 2013-2017 to provide a set of
7-day hindcasts (Figure 2). Streamflow hindcasts both from the ESP and the
hydrometerological approach (TIGGE-ECMWF/CSSPv2) were corrected by
matching monthly mean streamflow observations to remove the biases, and the
hindcast experiments were termed as ESP-Hydro and Meteo-Hydro (Table 2).
*2.2.4 LSTM streamflow forecast*
LSTM is a type of recurrent neural network model which learns from sequential
data. The input of the LSTM model includes forecast interval streamflow at the
specified forecast step obtained from TIGGE-ECMWF/CSSPv2, historical upstream
streamflow observation, and historical streamflow observation at Yantan hydrological
gauge. The network was trained on sequences of April to September in 2013-2017,
with six historical streamflow observations and one forecast interval streamflow to
predict the total streamflow at each forecast time step (Figure 2). The LSTM was
calibrated through a cross validation method, by leaving the target year out.
Before calibration, all input and output variables were normalized as follows:
$$\mathbf{q_0} = \frac{(\mathbf{q}-\mathbf{q_{min}})}{(\mathbf{q_{max}}-\mathbf{q_{min}})},\tag{7}$$

where $\mathbf{q}$, $\mathbf{q_{max}}$ and $\mathbf{q_{min}}$ are the input variable, the maximum and minimum of the
sequence of the variable. The hindcast experiment was termed as
Meteo-Hydro-LSTM (Table 2). In addition, we also tried an LSTM streamflow



forecast approach which only uses 6-hr historical streamflow data as inputs, and the
experiment was termed as LSTM (Table 2).
*2.3 Evaluation Method*
The root-mean squared error (RMSE) was used to evaluate the deterministic
forecast, i.e., the ensemble means of 51 (ECMWF) or 40 (ESP) forecast members. To
evaluate probabilistic forecasts, the Continuous Ranked Probability Score (CRPS)
was calculated as follows:
$$CRPS = \int_{-\infty}^{\infty} [F(y) - F_o(y)]^2, \tag{1}$$

where
$$F_o(y) = \begin{cases} 0, & y < observed\ value \\ 1, & y \geq observed\ value \end{cases} \tag{2}$$

is a cumulative-probability step function that jumps from 0 to 1 at the point where the
forecast variable $y$ equals the observation. The CRPS has a negative orientation
(smaller values are better), and it rewards concentration of probability around the step
function located at the observed value (Wilks, 2005). The skill score for deterministic
forecast was calculated as
$$SS_{RMSE} = \frac{RMSE - RMSE_{ref}}{0 - RMSE_{ref}} = 1 - \frac{RMSE}{RMSE_{ref}} \quad . \tag{3}$$

The skill score for probabilistic forecast (CRPSS) could be calculated similarly based
the CRPS.
**3. Results**





*3.1 Evaluation of CSSP calibration*

The employed CSSPv2 model is a fully distributed hydrological model and the streamflow is calculated through a process of converting gridded rainfall into runoff and a process of runoff routing. Figure 3 shows the runoff calibration results by calculating the NSE of monthly runoff simulations compared with observed gridded monthly runoff. After calibrating the CSSPv2 runoff model, the NSE of all grids are above 0, which indicates that the runoff simulation results in all grids are more reliable than the climatology method. In addition, grids distributed in the downstream region have better NSE than the upstream grids. The NSE values of the grids in the southern part are greater than 0.5, which accounts for two thirds of the interval basin area.

Figures 4 and 5 show the results after the calibration of the routing model, where time series of CSSPv2-simulated streamflow are compared against observed streamflow at Yantan hydrological gauge. Figure 4 shows the daily and monthly streamflow simulation results. The monthly result (Fig. 4f) shows that the simulated streamflow closely follows the observed streamflow, and the NSE is 0.96. The daily streamflow simulations during flood seasons (Figs. 4a-4e) also show a good performance, and the NSE is 0.92. During June and July in years of 2014, 2015 and 2017, the CSSPv2 model underestimated the daily streamflow with a maximum of 1104 $m^3$/s and an average of 334 $m^3$/s (Figs. 4b, 4c, 4e). In years of 2013 and 2016, the difference between observed and simulated streamflow is relatively small, and the average difference is 96 $m^3$/s (Figs. 4a, 4d).



Figure 5 shows the hourly streamflow simulation results for a few flood events.
Figure 5a shows that the CSSPv2 model can accurately simulate the streamflow
response to a rainfall event after a dry period. Figures 5b-5d show that for
instantaneous heavy rainfall events, the CSSPv2 model over-predicted the water loss
during recession period. Figures 5e-5f show that for continuous rainfall events, the
simulated streamflow has a larger fluctuation than observation. The simulated
streamflow is also smoother than observation. Nevertheless, the NSE for the hourly
streamflow simulation is 0.61, which suggests that CSSPv2 has an acceptable
performance at hourly time scale.
*3.2 Bias correction of TIGGE-ECMWF meteorological forecasts*
The resolution of TIGGE-ECMWF grid data is 0.25°, so the data was
interpolated to 5km grid to drive the CSSPv2 model. Figure 6 shows the correlation
coefficient and RMSE of TIGGE-ECMWF precipitation and temperature forecasts as
compared against observations, either before or after bias correction. The 51-ensemble
mean precipitation and temperature (the red dashed lines) shows better performance
than the best ensemble members (the green dashed lines), with an average RMSE
reduction of 3.66 mm/day and average correlation increase of 0.04 for precipitation,
and average RMSE reduction of 0.1K and average correlation increase of 0.03 for
temperature. After bias correction, the 51-ensemble means still perform better than
best ensemble members. Compared with ensemble mean results before bias correction,
the RMSE reduced by 0.23 mm/day for the bias-corrected precipitation, and reduced
by 1K for the bias-corrected surface air temperature. For the bias-corrected ensemble
mean results, the average RMSE and correlation are 14.6 mm/day and 0.44 for
precipitation, and 1.25 K and 0.87 for surface air temperature.
*3.3 Comparison between ESP-Hydro and Meteo-Hydro streamflow forecast*

274       Figure 7 presents the variations of RMSE and CRPS for ESP-Hydro and

Meteo-Hydro hourly streamflow forecast at Yantan hydrological gauge. For
probabilistic forecast, Figure 7a shows that the CRPS for Meteo-Hydro streamflow
forecast ranges from 160 to 230 while the CRPS for ESP-Hydro streamflow forecast
ranges from 183 to 250. The Meteo-Hydro approach performs better than ESP-Hydro
with lower CRPS at all lead times, with an average of 10% improvement in CRPSS
(Figure 7c). For deterministic forecast, Figure 7b shows that the RMSE for
Meteo-Hydro streamflow forecast ranges from 250 to 350 $m^3$/s, while the RMSE for
ESP-Hydro streamflow forecast ranges from 250 to 390 $m^3$/s. The Meteo-Hydro
approach also performs better than ESP-Hydro with lower RMSE at all lead times
especially after 3 days, with the average reduction of RMSE reaching 6% (Figure 7d).

285       Figure 7 also shows that both forecast skills have a similar diurnal cycle, where

RMSE and CRPS reach their peaks around 00UTC and drop to their lows at 06UTC.
Figure 8 shows the diurnal cycle of model employed variables, which are observed
catchment mean rainfall, observed streamflow at Yantan and Longtan hydrological
gauges, to explain the diurnal cycle of ESP-Hydro and Meteo-Hydro forecasting skills.
These three input variables show different diurnal patterns. The observed rainfall
starts to rise at 00UTC and reaches its maximum at 06UTC. The observed streamflow
at Yantan hydrological gauge drops to its minimum at 12UTC and rises to its



maximum at 00UTC. The streamflow from upstream Longtan hydrological gauge
starts to drop at 00UTC and reaches its minimum at 06UTC. After comparing these
diurnal cycles with the cycle of forecast skill, it is found that the forecast skill
decreases when the upstream Longtan outflow starts to decrease, and the precipitation
starts to increase. When the upstream Longtan outflow increases and the precipitation
starts to decrease (after 06UTC), the forecast skill rises. Such information indicates
that the hydrological model performs worse in the case of heavier rainfall event, and
the decrease of upstream outflow may amplify such degradation when the portion of
interval rainfall-runoff increased.
*3.4 Meteo-Hydro-LSTM streamflow forecast*
Machine learning methods can recognize patterns hidden in input data and can
simulate or predict streamflow without explicit descriptions of the underlying physical
processes. Figure 9 shows the RMSE of Meteo-Hydro-LSTM streamflow forecast
using the ensemble mean hydrological forecast as described in the section above, and
the past 6-hour observed streamflow of Yantan hydrological gauge as input.
Compared with Meteo-Hydro and ESP-Hydro approach, applying LSTM model can
further decrease the RMSE within the first 72 hours. The RMSE of
Meteo-Hydro-LSTM approach ranges from 205 to 363 $m^3$/s during these three days,
suggesting an average of 6% improvement against Meteo-Hydro approach.
Figure 9 also shows the RMSE of LSTM streamflow forecast only using the past
6-hour observed streamflow of Yantan hydrological gauge as input. Without using the





physical model forecast, RMSE is improved only when the lead time is less than 1 day.
And the performance of LSTM is far worse than Meteo-Hydro streamflow forecast
when lead time is more than 2 days.

317        Figure    10    shows    several    examples    of    streamflow    forecasts    by

Meteo-Hydro-LSTM approach and Meteo-Hydro approaches to show the forecast
improvements in details. The Meteo-Hydro-LSTM approach reduced the flood peak
value and the water loss during flood recession period compared with Meteo-Hydro
streamflow forecast approach, which improves the streamflow prediction for most
cases (Figs. 10b-10f). However, when the upstream reservoir's flood operation is
triggered    by    continuous    heavy    rain,    the    Meteo-Hydro    may    underpredict    the
streamflow. With    the    LSTM    model    further    decreases    the    streamflow,    the
Meteo-Hydro-LSTM method can end up with worsening the streamflow forecast,
which means the machine learning method may improve forecasts when trained in
different flood operating situations (Figure 10a).
**4. Conclusions**

329        In this study, we developed and evaluated a streamflow forecasting framework

by coupling meteorological forecasts with a land surface hydrological model (CSSPv2)
and a machine learning method (LSTM) over a cascade reservoir catchment using
hindcast data from 2013 to 2017. The monthly observed runoff was used to calibrate
the runoff generation module of the CSSPv2 model grid by grid, and the hourly
observed streamflow at Yantan hydrological gauge was used to calibrate the routing



module of the CSSPv2 model. Then, the bias-corrected TIGGE-ECMWF ensemble
forecasts were used to drive the CSSPv2 for streamflow forecasts, and the LSTM
model was used to correct the streamflow forecasts, resulted in an integrated
meteorological-hydrological-machine learning forecast framework.
With automatic offline calibration of the CSSPv2 model, and the NSE values are
0.96, 0.92 and 0.61 for streamflow simulations at the Yantan gauge at monthly, daily
and hourly time scales, respectively. The bias-corrected ensemble mean
TIGGE-ECMWF forcings which perform the best among all ensemble members,
show average RMSE and correlation of 14.6 mm/day and a 0.44 for precipitation
forecasts, and 1.3 K and 0.87 for surface air temperature forecasts. By comparing with
the hourly observed streamflow, the integrated hydrometeorological forecast approach
(Meteo-Hydro) increases the probabilistic and deterministic forecast skill against the
initial condition-based approach (ESP-Hydro) by 10% (CRPSS) and 6% (RMSE skill
score), respectively .
Adding LSTM model to the hydrometeorological forecast (Meteo-Hydro-LSTM)
can further reduce the forecast error. Within the first 72 hours, LSTM can improve the
forecast skill with a maximum of 25% and an average of 6%. However, if we do not
use the streamflow predicted by Meteo-Hydro, the error from the LSTM increases
rapidly after 24 hours, and the historical data-based LSTM method performs worse
than the Meteo-Hydro method.



**Acknowledgement.** This work was supported by National Key R&D Program of
China (2018YFA0606002), and National Natural Science Foundation of China

358 (41875105).


**Data availability.** The TIGGE-ECMWF hindcast data can be downloaded from
https://apps.ecmwf.int/datasets/data/tigge/levtype=sfc/type=pf/ (Parsons et al., 2017),
the in-situ observations and simulation data are available upon request.



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





**Table 1.** Information of hydrological gauges.

| Gauge | Longitude (°E) | Latitude (°N) | Drainage area (km$^2$) |
|---|---|---|---|
| Longtan | 107.09 | 25.00 | - |
| Yantan | 107.50 | 24.11 | 5950 (orange area in Fig. 1) |
| Luofu | 107.36 | 24.90 | 800 (green area in Fig. 1) |
| Jiazhuan | 107.12 | 24.21 | 2150 (purple area in Fig. 1) |





**Table 2.** Experimental design in this study.

| Experiments | Description |
| --- | --- |
| ESP-Hydro | Using CSSPv2 land surface hydrological model driven by randomly-sampled historical meteorological forcings |
| Meteo-Hydro | Using CSSPv2 model driven by bias-corrected TIGGE-ECMWF hindcast meteorological forcings |
| Meteo-Hydro-LSTM | Using LSTM model to correct streamflow from Meteo-Hydro hindcast |
| LSTM | Using LSTM model to forecast streamflow based on observation only |







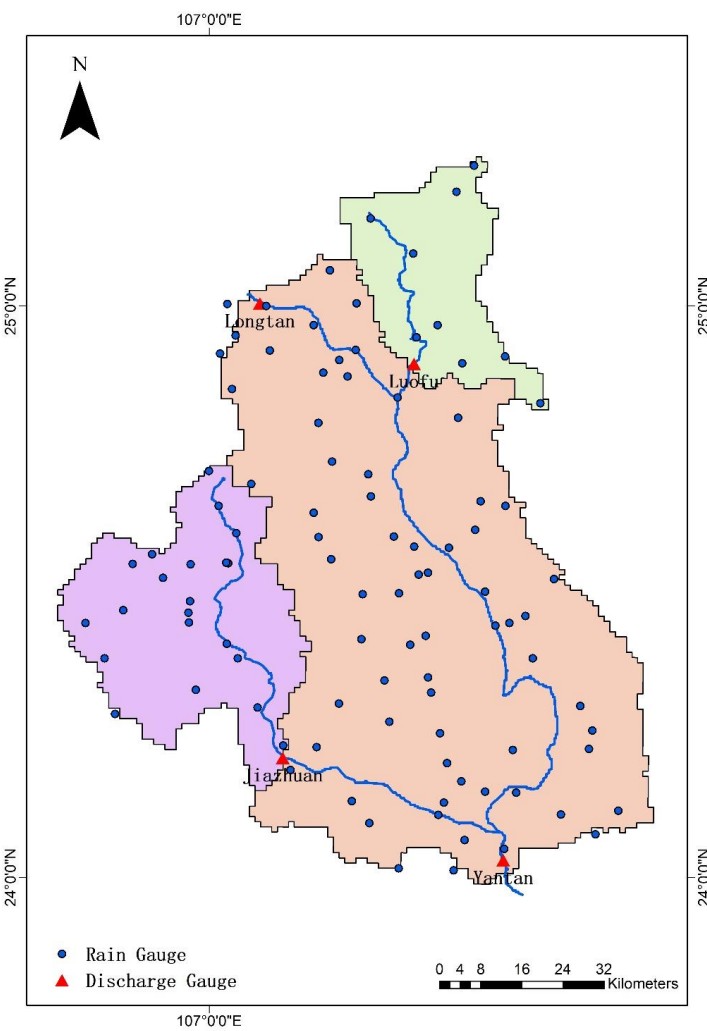


**Figure 1.** Locations of discharge gauges and rain gauges over the Yantan basin.






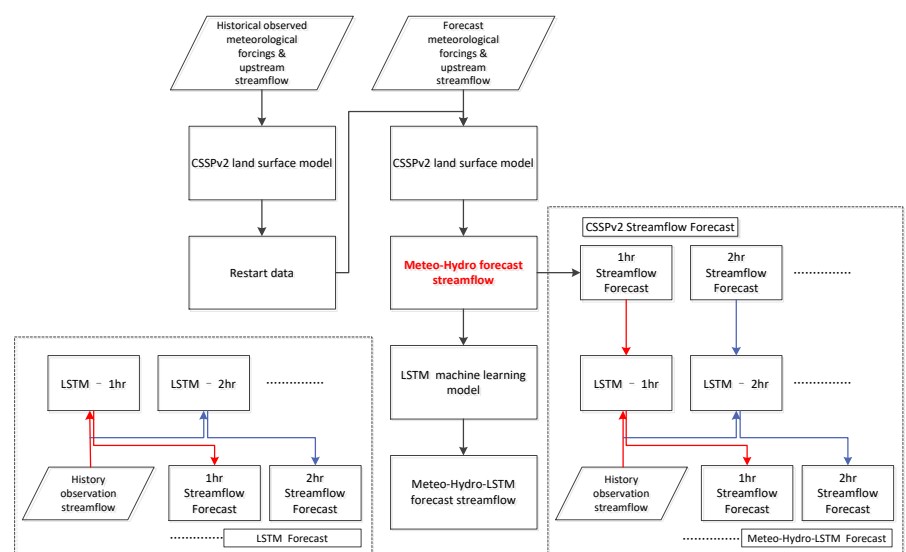


**Figure 2.** A diagram for the integrated hydrometeorological and machine learning

streamflow prediction.






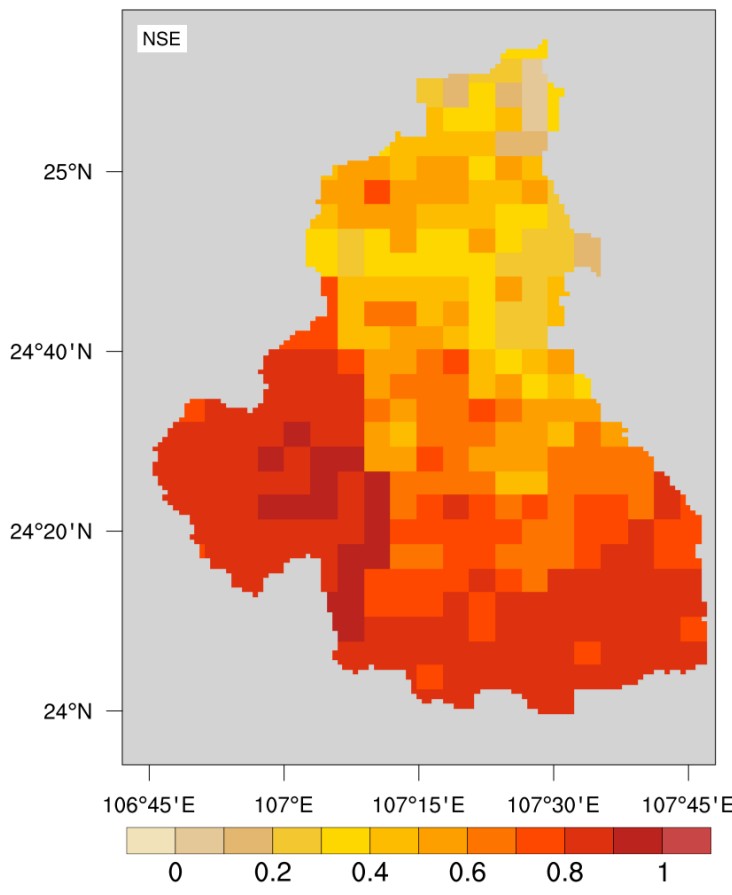


**Figure 3.** Nash-Sutcliff efficiency coefficients for the calibrated grid runoff simulation

from CSSPv2.


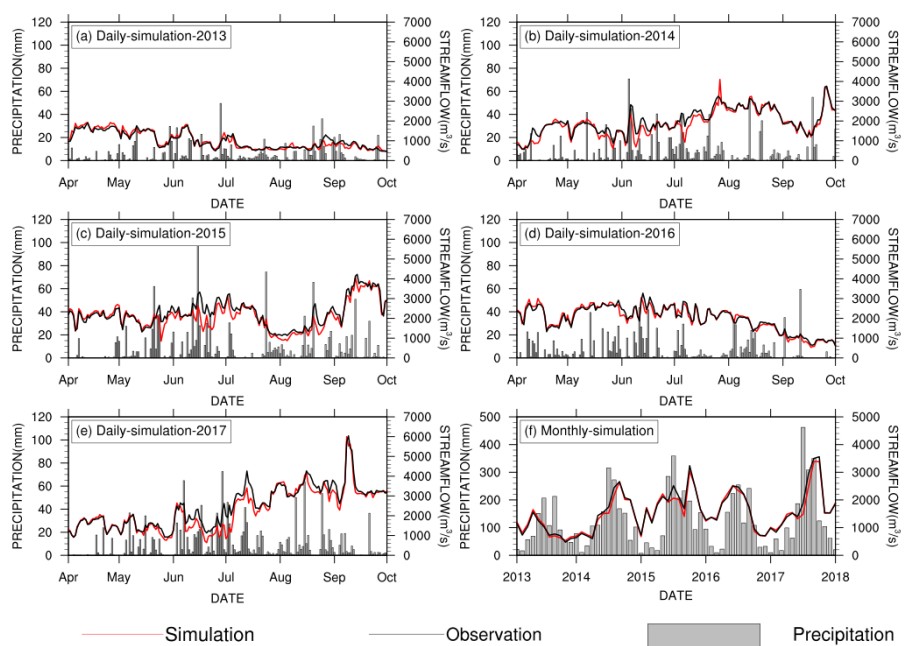

**Figure 4.** Evaluation of streamflow simulations at Yantan gauge. The black and red

lines are observed and simulated streamflow. (a)-(e) are for daily streamflow, and (f)

is for monthly streamflow. The gray bars represent daily (or monthly) precipitation.










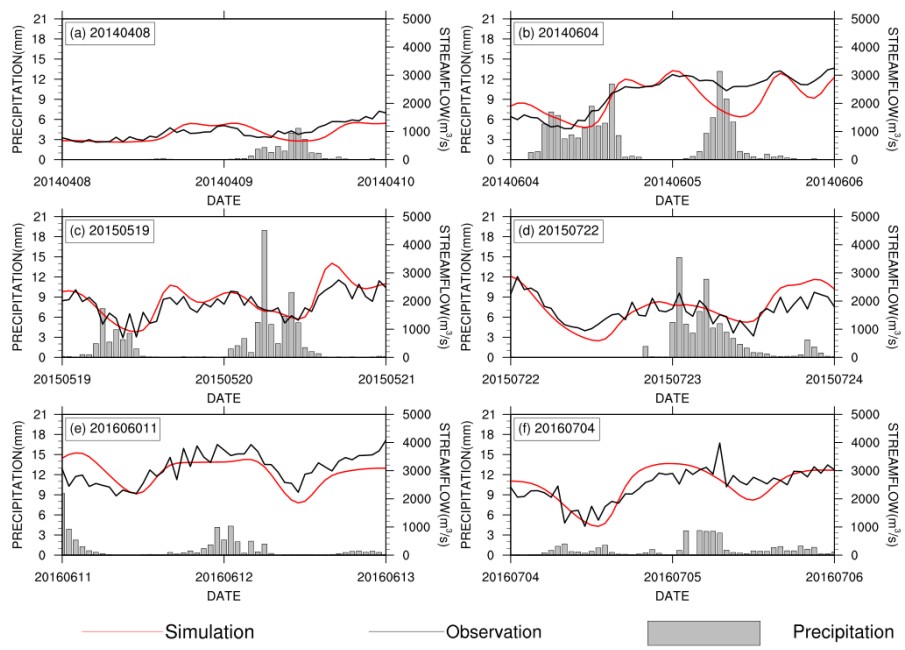


**Figure 5.**    The same as Figure 4, but for the evaluation of hourly streamflow
simulations at Yantan gauge.




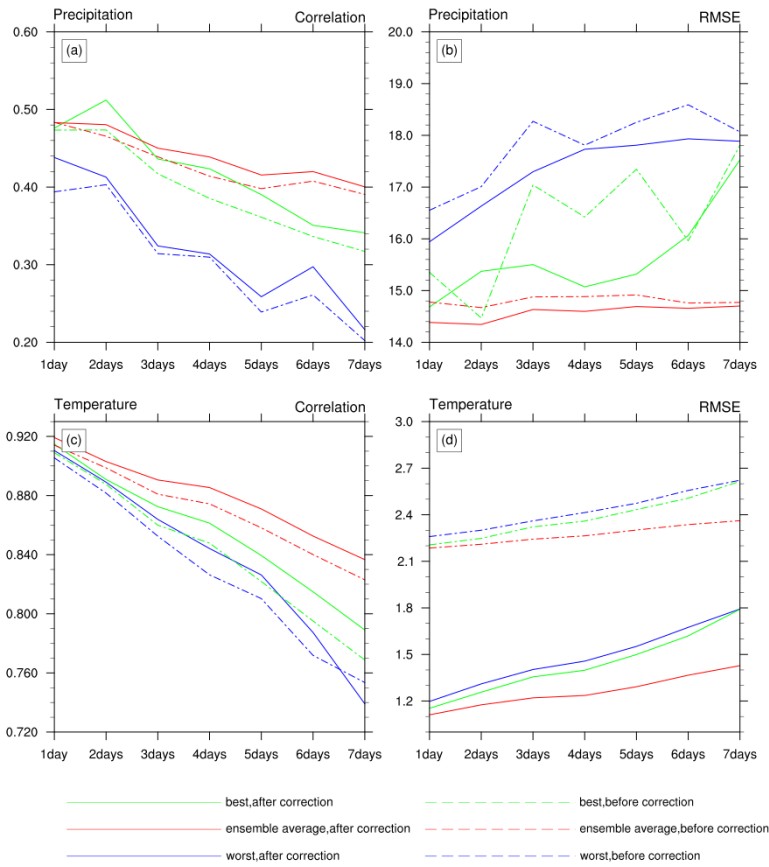


**Figure 6.** Evaluation of precipitation and temperature hindcasts from TIGGE-ECMWF. The red and blue lines represent the best and worst results among 51 TIGGE-ECMWF ensemble members respectively, and the green lines represent the results for the ensemble means of 51 members. Solid and dashed lines represent the results after and before bias corrections, respectively.




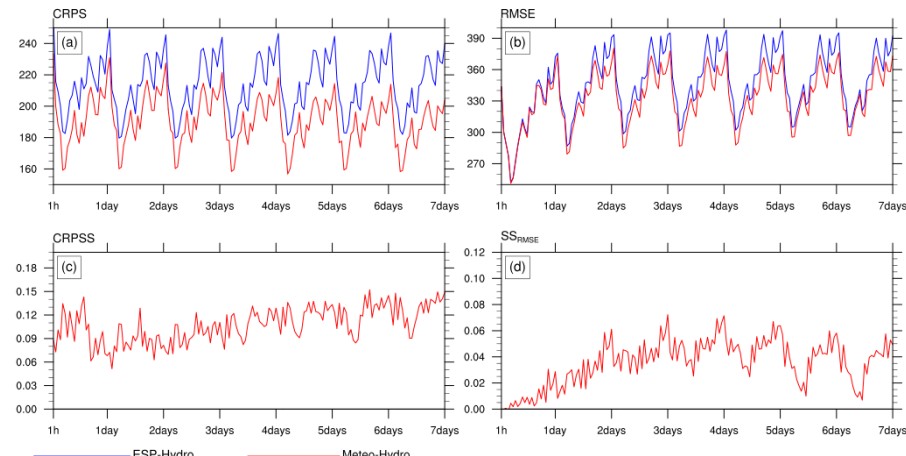


**Figure 7.** (a) Continuous Ranked Probability Score (CRPS) and (b) Root Mean
Squared Error (RMSE) for daily streamflow ensemble forecasts at Yantan gauge. (c)
and (d) are the skill score in terms of CRPS and RMSE for Meteo+Hydro, where
ESP+Hydro is used as reference forecast.






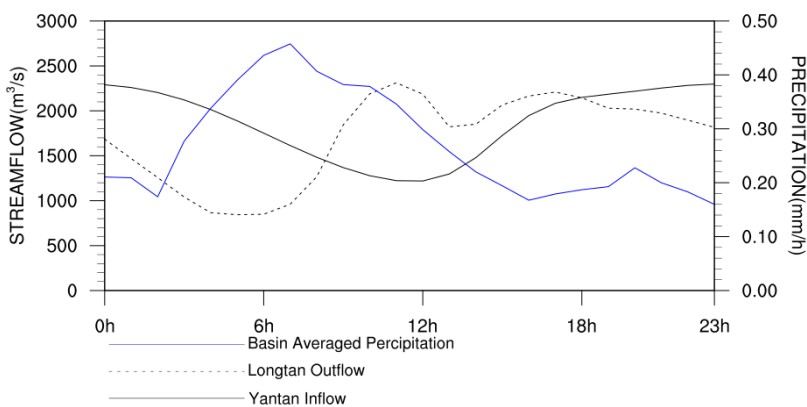


**Figure 8**. Diurnal cycle of Longtan outflow (m$^3$/s; dashed black line), Yantan inflow

(m$^3$/s; solid black line) and basin-averaged precipitation (mm/h; blue line).


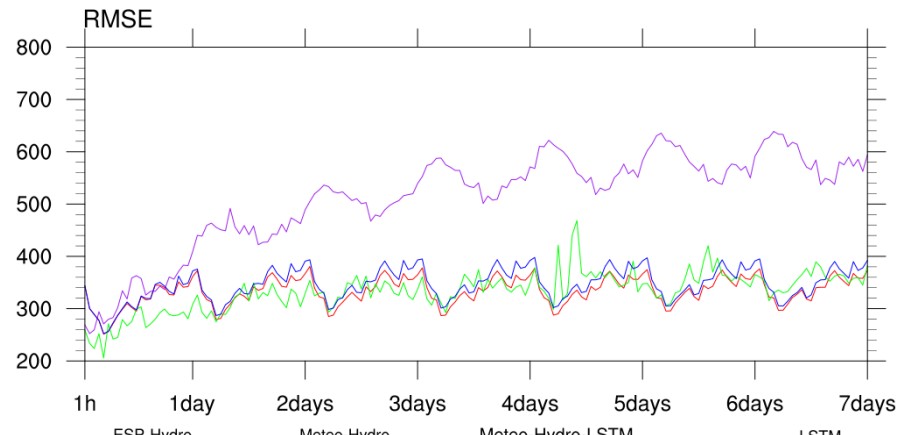


**Figure 9.** RMSE (m³/s) for hourly streamflow hindcasts from four forecast
approaches. The green line represents the Meteo+Hydro+LSTM forecast, the red line
represents the Meteo+Hydro forecast, the blue line represent the ESP+Hydro forecast,
and the purple line represents the LSTM forecast based on historical streamflow
observation alone.



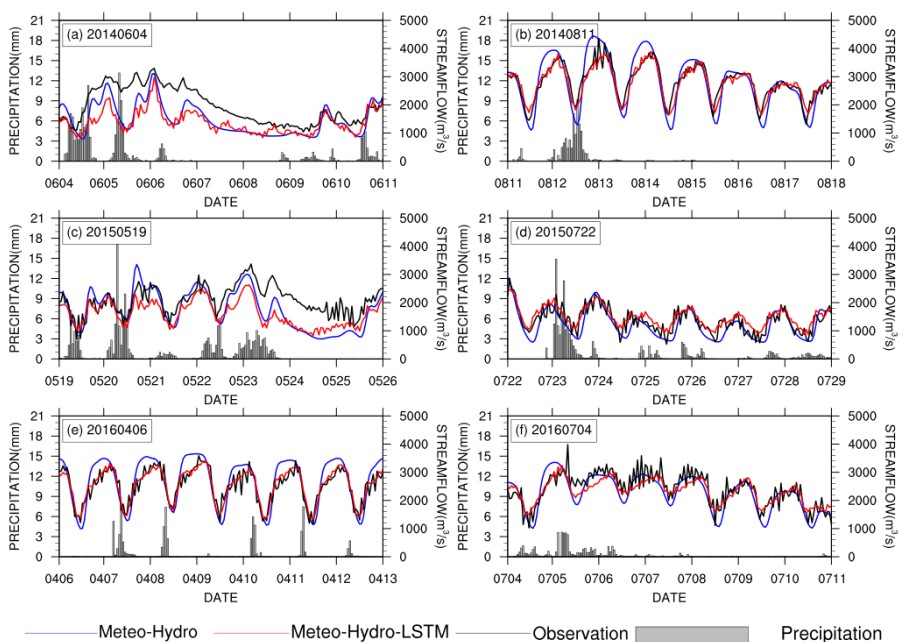


Figure 10. Evaluation of the forecast approaches for a few flooding events. The black

lines are observed streamflow from Yantan hydrological gauge, the blue lines are the

Meteo+Hydro ensemble mean streamflow forecast, and the red lines are the

Meteo+Hydro+LSTM forecast streamflow by using Meteo+Hydro ensemble mean

forecast with LSTM. The gray bars represent hourly precipitation averaged over the

basin.