# Peer review of "Ensemble streamflow forecasting over a cascade reservoir catchment with integrated hydrometeorological modeling and machine learning"

_Hydrology and Earth System Sciences, 2021_

## Author Response (AR1)

Xing Yuan
Professor
School of Hydrology and Water Resources
Nanjing University of Information Science and Technology
No.219, Ningliu Road, Nanjing 210044, Jiangsu, China
Email: xyuan@nuist.edu.cn
Tel: +86-025-58699958
https://orcid.org/0000-0001-6983-7368
October 12, 2021

Prof. Bob Su

Editor

Hydrology and Earth System Sciences

Re: hess-2021-393

Dear Prof. Su,

Regarding your decision letter on our manuscript entitled "Ensemble streamflow forecasting over a cascade reservoir catchment with integrated hydrometeorological modeling and machine learning" (hess-2021-393), we have now carefully considered the reviewers' comments and incorporated them into the manuscript to the extent possible. We hope that you find the revised manuscript and the response acceptable to *Hydrology and Earth System Sciences*. The detailed responses to the comments are attached.

We appreciate the effort you spent to process the manuscript and look forward to hearing from you soon.

Sincerely yours,

Xing Yuan

**Response to the comments from Reviewer #1**

We are grateful to the reviewer for the constructive and careful review. We have incorporated the comments to the extent possible. The reviewer's comments are italicized and our responses immediately follow.

*The authors proposed an integrated ensemble prediction approach based on hydrometeorological modeling and machine learning for streamflow forecasting over a cascade reservoir catchment. The performance of the prediction with different model settings is compared. Results show the potential of the integrated hydro-meteorological and machine learning approach. Some comments are provided as follows and need to be addressed before the potential publication of this study.*

**Response:** We would like to thank the reviewer for the positive comments. Please see our responses below.

*1) Line 114: Please give the full name of CSSPv2*

**Response:** We have revised as "... a newly developed high-resolution land surface model, named as the Conjunctive Surface-Subsurface Process model version 2 (CSSPv2, Yuan et al., 2018), to provide runoff and streamflow forecasts ..." (L119-L120 in the tracked version of the revised manuscript)

*2) Lines 145-146: The authors calibrate the model based on the runoff at each grid (instead of using the streamflow at the control station). What is the motivation or advantage of this model calibration?*

**Response:** Thanks for your comments. The calibration can be divided into two steps: the calibration of parameters for rainfall-runoff generation process for each grid, and the calibration of parameters for river routing process for the entire catchment. We have clarified the motivation in the revised manuscript as follows:

"The gridded runoff was used to calibrate the CSSPv2 model at each grid (Yuan et al., 2016), which would generate distributed model parameters that are different within the catchment to better represent the heterogeneity of the rainfall-runoff processes." (L155-L158)

The detailed description of model calibration can be found in section 2.2.3.

*3) Line 217, equation (2); Here "y>observation" is assigned "1". In other words, no matter how high the simulation is (if higher than the observation), it will result in a low CRPS, right? By the way, is this commonly used in previous studies?*

**Response:** We used the CRPS definition according to Wilks (2005). A low CRPS value means the forecast distribution yields a better concentration of probability around the step function located at the observed value. If forecast members got extremely higher or lower values than the observation, the CRPS will increase because $[F(y)-F_o(y)]^2$ will increase. The CRPS is widely used in evaluating the accuracy of forecasts, and a skill score of CRPS (i.e., CRPSS) is used to compare with a benchmark ensemble forecast.

*4) Lines 234-235: The simulation is better for the downstream. Is there any specific reason for this pattern?*

**Response:** Thanks for your comments. We have clarified in the revised manuscript as follows:

"Higher NSE in the upstream part of Jiazhuan station (Figure 1) is due to more humid climate (not shown), where hydrological models usually have better performance over wetter areas. For the downstream areas with less precipitation, the higher NSE is related to the higher percentage of sand in the soil (not shown). Under the same meteorological conditions, there is higher hydraulic conductivity with higher sand content (Wang et al., 2016), and it yields less runoff under infiltration excess, which is more suitable for the saturation excess-based runoff generation for the CSSPv2 model (Yuan et al., 2018b)." (L283-L290)

Figures R1 and R2 show the distributions of precipitation and sand percentage.

[Figure]

**Figure R1**. The spatial distribution of average yearly precipitation (mm).

[Figure]

**Figure R2**. Spatial distribution of sand proportion.

*5) Lines 276-278 (Figure 7): For the CRPS and RMSE for the lead time of seven days, there is a strong cycle in the performance. What is the reason for the strong cycle? In addition, from the RMSE, we see low RMSE values for the lead time within 1-day, and relatively high values after 1-day. However, for the CRPS, such a pattern does not exist. In addition, for the lead time beyond 1-day, the variation of CRPS and RMSE does not seem to depend on the lead time (prediction performance generally degraded for longer lead time, right?). Please clarity.*

**Response:** Thanks for the comments. The diurnal cycles of CRPS and RMSE are associated with the diurnal cycle of upstream water release from Longtan station, and the diurnal cycle of catchment-averaged precipitation. Please see Figure 8 and related text (L345-L359)

About the CRPS pattern, we are sorry to inform that there were several mistakes during programming. The revised figure has been now portraited as below, which looks normal now for both CRPS and RMSE, where the error increases as the forecast lead increases.

[Figure]

**Figure 7.** (a) Continuous Ranked Probability Score (CRPS) and (b) Root Mean Squared Error (RMSE) for daily streamflow ensemble forecasts at Yantan gauge. (c) and (d) are the skill score in terms of CRPS and RMSE for Meteo+Hydro, where ESP+Hydro is used as reference forecast.

*6) It is hard to read the station name in this figure. Please improve it.*

**Response:** Thanks for the comment. We have redrawn Figure 1 as follows:

[Figure]

**Figure 1.** Locations of discharge gauges and rain gauges over the Yantan catchment.

*7) Are reservoir data used in the simulation or incorporated in the model?*

**Response:** The upstream reservoir (Longtan) discharge data was used as model's upstream inputs. We have clarified this caveat in the discussion as follows:

"This study mainly focused on exploring the added values of meteorology-hydrology coupled forecast and LSTM forecast in a non-closed catchment, so the forecast uncertainty from upstream outflow was ignored by using the observed outflow. In the

future, the upstream outflow forecast is planned to include, but this requires the development of upstream hydrometeorological forecast capability, as well as the reservoir regulation forecast that is very challenging. The artificial intelligence (AI) techniques are expected to complement the physical model for reservoir regulation forecast." (L415-L422)

**Reference:**

Wilks, D. S.: Statistical Methods in the Atmospheric Sciences, Volume 91, Second Edition International Geophysics, 2005.

Wang, Y., Fan, J., Cao, L., et al.: Infiltration and Runoff Generation Under Various Cropping Patterns in the Red Soil Region of China. Land. Degrad. Dev.  27(1), 83-91. https://doi.org/10.1002/ldr.2460, 2016.

Yuan, X., Ji, P., Wang, L., Liang, X. Z., Yang, K., Ye, A., et al.: High‐resolution land surface modeling of hydrological changes over the sanjiangyuan region in the eastern tibetan plateau: 1. model development and evaluation. J. Adv. Model Earth Syst. https://doi.org/10.1029/2018MS001412, 2018b.

**Response to the comments from Reviewer #2**

We are grateful to the reviewer for the constructive and careful review. The constructive suggestions have helped improved our manuscript. The reviewer's comments are italicized and our responses immediately follow.

*Reservoirs represent an important but difficult issue for hydrological modelling. This paper presents a method for ensemble streamflow forecasting at the hourly timescale considering the effects of cascade reservoirs. The method makes use of TIGGE-ECMWF meteorological forecasts, CSSPv2 land surface model and LSTM deep learning model. Through the case study of a reservoir in China, the method is shown to reduce probabilistic and deterministic forecast errors. In general, the paper is well-written with results clearly presented.*

*There are five comments for further improvements of the paper.*

**Response:** We would like to thank the reviewer for the positive comments. Please see our responses below.

*1) First of all, more details on the contribution of this paper can be added. As is illustrated in the introduction, the proposed method is built upon the CSSPv2 land surface model (Yuan et al., 2018). What are the limitations of the previous model? Can the limitations be illustrated through some diagnostic plots? Such analysis would make the contribution of this paper more convincing.*

**Response:** Thanks for your comments. In this study, we combined the newly developed CSSPv2 land surface hydrological model with ECMWF meteorological forecasts and the LSTM machine learning model to develop a Meteo-Hydro-LSTM forecasting framework for flooding forecasts over a cascading reservoir catchment. We have clarified in the revised manuscript as follows:

"In this study, we combine the machine learning with hydrometeorological approach for hourly streamflow forecast over a cascade reservoir catchment located in southwestern China. We use the meteorological hindcast data from European Centre for Medium-Range Weather Forecasts (ECMWF) model that participated in the

THORPEX Interactive Grand Global Ensemble (TIGGE) project to drive a newly developed high-resolution land surface model, named as the Conjunctive Surface-Subsurface Process model version 2 (CSSPv2, Yuan et al., 2018), to provide runoff and streamflow forecasts, and correct the forecasts via LSTM model. We aim to improving flood forecast over the cascade reservoir catchment by integrating meteorological forecast, hydrological modeling and machine learning." (L114-125 in the tracked version of the revised manuscript)

*2) Second, the method is demonstrated for one reservoir. In the meantime, the "study area" section illustrates that there are ten cascade reservoirs in the Hongshuihe hydropower base. Is it possible to select another 2-3 reservoirs to show the robustness of the proposed method? It is noted that the additional case study reservoirs can be elsewhere and are not necessarily located in the Hongshuihe region.*

**Response:** Thanks for your suggestions. Our ultimate goal is to develop a forecast system that can consider both upstream and downstream reservoirs. However, as the first step, we focus on assessing the added value of integrating meteorological forecast, hydrological modeling and machine learning in the flood forecasting. We have clarified this caveat in the discussion as follows:

"This study mainly focused on exploring the added values of meteorology-hydrology coupled forecast and LSTM forecast in a non-closed catchment, so the forecast uncertainty from upstream outflow was ignored by using the observed outflow. In the future, the upstream outflow forecast is planned to include, but this requires the development of upstream hydrometeorological forecast capability, as well as the reservoir regulation forecast that is very challenging. The artificial intelligence (AI) techniques are expected to complement the physical model for reservoir regulation forecast." (L415-L422)

*3) Third, Figure 8 presents an interesting illustration of the time lag between Longtan outflow and Yantan inflow. This lag is largely due to the flowing distance between the*

*two reservoirs. Meanwhile, the section of methods does not tell how the river flow is considered in the method. Is it performed by routing or hydro-dynamic simulation? How are the parameters determined?*

**Response:** Thanks for your comments. The river flow was calculated by a routing model employed the kinetic wave equation as covariance function, which was solved via a Newton algorithm (L186-L187). The parameters include slope, river density, roughness, width, and depth (L198-L199). These parameters were calibrated to match the hourly observed streamflow at Yantan hydrological gauge. We have clarified the routing model as follows:

"We used a high-resolution elevation database (hereafter referred to as DEM30) for sub-grid parameterization and figured out the initial values of these river channel parameters. We first extracted the slope angle and the natural river flow path from DEM30, and then identified the accurate river network using a drainage area threshold of 0.18 km$^2$. River density and bed slope values for each 5km grid were calculated as:

$$rivden = \sum l / A \qquad (1)$$

$$bedslp = mean(\tan(\beta)) \qquad (2)$$

where *rivden* is the river density (km/km$^2$), *bedslp* is the river channel bed slope (unitless), $A$ is the area of a 5km grid (km$^2$), $\Sigma l$ is the total river channel length (m) within the grid, $\beta$ is the slope angle (radian) for each river segment located in the grid.

Other river channel parameters were estimated by empirical formulas (Getirana et al., 2012; Luo et al., 2017) as follows:

$$W = 1.956 \times A_{acc}^{0.413} \qquad (3)$$

$$H = 0.245 \times A_{acc}^{0.342} \qquad (4)$$

$$n = 0.03 + (0.05 - 0.03) \frac{H_{max} - H}{H_{max} - H_{min}} \qquad (5)$$

where *W, H* and *n* are river width (m), depth (m) and roughness (unitless) for each 5km grid; $A_{acc}$ means the upstream drainage area (km$^2$); $H_{max}$ and $H_{min}$ refer to the maximum and minimum values of river depth calculated by Eq. (4).

Through a trial-and-error procedure, we calibrated these river channel parameters to match the simulated streamflow with observed hourly records at Yantan hydrological gauge." (L200-L221)

4) *Fourth, lead time plays an important part in forecast verification as forecast skill tends to decrease with the increase lead time. Meanwhile, the simulations shown in Figures 4 and 5 seem to have nothing to do with lead time. Please present some plots of ensemble forecasts at different lead times*

**Response:** Sorry for the confusion we made in the manuscript. The simulations shown in Figures 4 and 5 are driven by the observed meteorological forcings in order to evaluate the performance of the CSSPv2 land surface hydrological model. They are not "real" forecasts. The performance of the ensemble streamflow forecasts are shown via CRPS plots in Figure 7. We have clarified in the revised manuscript as follows: "Figures 4 and 5 show the results after the calibration of the routing model, where CSSPv2 is driven by observed meteorological forcings to provide streamflow simulations and compare against observed streamflow at Yantan hydrological gauge." (L291-293)

5) *Fifth, CRPS in Figure 7 exhibits some diurnal circle that can relates to the diurnal circle of reservoir inflow/outflow in Figure 8. This result may be due to the setting of the LSTM deep learning model. When preparing streamflow data for LSTM, has the mean been subtracted? Are alternative settings, e.g., subtracting the mean or not, tested for LSTM?*

**Response:** Thanks for your comments. The results in Figure 7 didn't include the LSTM deep learning model, while they are based on CSSPv2 streamflow forecasts driven by TIGGE-ECMWF meteorological forecasts or climatological forecasts (i.e., ESP). The diurnal cycle of CRPS is related to the upstream water release from Longtan station, and the diurnal cycle of catchment-averaged precipitation. Please see Figure 8 and related text (L345-L359).

When performing LSTM corrections, we didn't subtract the mean streamflow, but normalized the streamflow data with the maximum and minimum streamflow. Please see section 2.2.4 for details.

*6)  The location map can be improved by illustrating all the reservoirs in the Hongshuihe hydropower base. In addition, the location of the Hongshuihe hydropower base in China can be presented by using an inset plot.*

**Response:** Thanks for the suggestion. We have redrawn Figure 1 as below.

[Figure]

**Figure 1.** Locations of discharge gauges and rain gauges over the Yantan basin.

*7)  In Table 1, please illustrate the year/month range and time step for the hydrological dataset.*

**Response:** Thanks for the suggestion. We have revised as below.

**Table 1.** Information of hydrological datasets

| Dataset | Time Range | Time step |
|---|---|---|
| Rain Gauge Observation Forcing | 2013/1/1 ~ 2017/12/31 | Hourly |
| Longtan & Yantan Discharge Gauge Streamflow data | 2013/1/1 ~ 2017/12/31 | Hourly |
| Jiazhuan & Luofu Discharge Gauge Streamflow data | 2013/4/1 ~ 2017/9/30 | Daily |
| TIGGE-ECMWF Forecast Forcing | 2013/4/1 ~ 2017/9/30 | Hourly |

**References:**

Getirana, A. C. V., Boone, A., Yamazaki, D., Decharme, B., Papa, F., and Mognard, N.: The Hydrological Modeling and Analysis Platform (HyMAP): Evaluation in the Amazon Basin, J. Hydrometeorol., 13, 1641–1665, https://doi.org/10.1175/JHM-D-12-021.1, 2012.

Luo, X., Li, H. Y., Ruby, L. L., Tesfa, T. K., Augusto, G., & Fabrice, P., et al.: Modeling surface water dynamics in the amazon basin using mosart-inundation v1.0: impacts of geomorphological parameters and river flow representation. Geosci. Model. Dev., 10(3), 1-42. https://doi.org/10.5194/gmd-10-1233-2017 , 2017.

**Response to the comments from Reviewer #3**

We are grateful to the reviewer for the constructive and careful review. We have incorporated the comments to the extent possible. The reviewer's comments are italicized and our responses immediately follow.

*The manuscript "Ensemble streamflow forecasting over a cascade reservoir catchment with integrated hydrometeorological modeling and machine learning" integrate the meteorological forecast, land surface hydrological model simulation and machine learning to forecast hourly streamflow over the Yantan catchment, and the results show that the flood forecast has been significantly improved. This work is very meaningful and the paper has been well-written. I therefore recommend this paper resubmitted after minor revisions. My comments are listed as follows:*

**Response:** We would like to thank the reviewer for the positive comments. Please see our responses below.

*1) The number of ESP members is 40, while the number of members used from TIGGE-ECMWF is 51, which is not the same as ESP, Will the evaluation results over predicted due to the number of members used from TIGGE is larger?*

**Response:** Thanks for your comments. First, the meteorological forcings from the same date as the hindcast start date to the next 9 days of each year (excluding the target year) were selected as the ESP forcing in this study. Although more members could be added to ESP, it will increase the overlap between different hindcasts due to limited historical samples. Second, larger sample of ESP means that more days far from the start date would be included, which may degrade the performance of ESP. Third, 40 members should be comparable to 51 members given the short hindcast periods.

*2) The upstream basin streamflow are used as CSSPv2 model inputs to provide the upstream inflow information. In the hindcast experiments, the upstream outflow inputs used are forecasts or observations? When lacking upstream outflow prediction, how does the system operate?*

**Response:** Thanks for your comments. In the hindcast experiments, the upstream outflow inputs used as inputs are observations to exclude the forecast uncertainty from upstream streamflow for the evaluation of the meteo-hydro-LSTM forecasting system. In the real-time forecasting system, we may use the upstream outflow from the previous day or the LSTM forecasts. We have clarified this caveat in the discussion as follows:

"This study mainly focused on exploring the added values of meteorology-hydrology coupled forecast and LSTM forecast in a non-closed catchment, so the forecast uncertainty from upstream outflow was ignored by using the observed outflow. In the future, the upstream outflow forecast is planned to include, but this requires the development of upstream hydrometeorological forecast capability, as well as the reservoir regulation forecast that is very challenging. The artificial intelligence (AI) techniques are expected to complement the physical model for reservoir regulation forecast." (L415-L422 in the tracked version of the revised manuscript)

*3) Is it possible to forecast the rainfall-streamflow using meteorological forcing from a closed watershed controlled by the Yantan station and correct it by LSTM instead?*

**Response:** Thanks for your comments. Yes, it is possible. But besides meteorological forecasts, the reservoir regulations should be incorporated or predicted, which is very challenging due to limited data and human interventions. Moreover, the motivation of this study is mentioned in the abstract as follows:

"For a cascade reservoir catchment, the discharge of the upstream reservoir contributes to an important part of the streamflow over the downstream areas, which makes it tremendously hard to explore the added value of meteorological forecasts. Here, we integrate the meteorological forecast, land surface hydrological model simulation and machine learning to forecast hourly streamflow over the Yantan

catchment, where the streamflow is influenced both by the upstream reservoir water release and the rainfall-runoff processes within the catchment."

As a first step, here we evaluate the performance of this meteo-hydro-LSTM coupled system. Future studies would incorporate upstream forecasts through a physical-statistical hybrid approach.

*4) The upstream inputs are also essential to this forecast system, hope to see some evaluation about this element.*

**Response:** The upstream inputs used in this system are observations, please see our response to your comment #2.

*5) The calibration results evaluated by NSE shows a worse result in the upstream grids than downstream ones. Is it possible to improve the calibration results by increasing the Iteration times set in the SCE-UA methods?*

**Response:** Thanks for your comments. We have clarified in the revised manuscript as follows:

"Higher NSE in the upstream part of Jiazhuan station (Figure 1) is due to more humid climate (not shown), where hydrological models usually have better performance over wetter areas. For the downstream areas with less precipitation, the higher NSE is related to the higher percentage of sand in the soil (not shown). Under the same meteorological conditions, there is higher hydraulic conductivity with higher sand content (Wang et al., 2016), and it yields less runoff under infiltration excess, which is more suitable for the saturation excess-based runoff generation for the CSSPv2 model (Yuan et al., 2018b)." (L283-L290)

Figures R1 and R2 show the distributions of precipitation and sand percentage. According to Figure R3, most grids reach its best performance within 2000 iteration times.

[Figure]

**Figure R1**. The spatial distribution of average yearly precipitation (mm).

[Figure]

**Figure R2**. Spatial distribution of sand proportion.

[Figure]

**Figure R3.** Iteration times when best performance occurred.

**Reference:**

Wang, Y., Fan, J., Cao, L., et al.: Infiltration and Runoff Generation Under Various Cropping Patterns in the Red Soil Region of China. Land. Degrad. Dev. 27(1), 83-91. https://doi.org/10.1002/ldr.2460, 2016.

Yuan, X., Ji, P., Wang, L., Liang, X. Z., Yang, K., Ye, A., et al.: High‑resolution land surface modeling of hydrological changes over the sanjiangyuan region in the eastern tibetan plateau: 1. model development and evaluation. J. Adv. Model Earth Syst. https://doi.org/10.1029/2018MS001412, 2018b.

**Response to the comments from Reviewer #4**

We are grateful to the reviewer for the constructive and careful review. We have incorporated the comments to the extent possible. The reviewer's comments are italicized and our responses immediately follow.

*This is an interesting manuscript that is well structured and presented. The topic is worthy of publication in HESS. However, before publication please address the following comments.*

**Response:** We would like to thank the reviewer for the positive comments. Please see our responses below.

*1) Line 201: What is the full name of GR4J? Please check throughout (e.g., TIGGE, ECMWF, CMA, SCE-UA).*

**Response:** Thanks for your comments. We have now provided the full names as follows:

"For example, Humphrey et al. (2016) showed that their combined Bayesian artificial neural network with the modèle du Génie Rural à 4 paramètres Journalier (GR4J) approach outperforms the GR4J model in monthly streamflow forecasting." (L104-L105 in the tracked version of the revised manuscript).

"We use the meteorological hindcast data from European Centre for Medium-Range Weather Forecasts (ECMWF) model that participated in the THORPEX Interactive Grand Global Ensemble (TIGGE) project to drive a newly developed high-resolution land surface model, named as the Conjunctive Surface-Subsurface Process model version 2 (CSSPv2, Yuan et al., 2018b), to provide runoff and streamflow forecasts, and correct the forecasts via LSTM model." (L116-L123).

"… China Meteorological Administration (CMA)." (L162-L163).

"In this paper, we calibrated CSSPv2 model against monthly estimated runoff to simulate the natural hydrological processes by using the Shuffled Complex Evolution (SCE-UA) approach (Duan et al., 1994)." (L191-L193).

*2) Lines 179-182: Please list the values or ranges of the calibrated parameters in a table.*

**Response:** Thanks for your suggestion. We have made a new table as follows:

**Table 2.** Descriptions of calibrated parameters

| Parameters | Range |
| --- | --- |
| Maximum velocity of baseflow (mm/day) | 0.00000116 ~ 0.000579 |
| Fraction of maximum velocity of baseflow where non-linear baseflow begins | 0.001 ~ 0.99 |
| Fraction of maximum soil moisture where non-linear baseflow occurs | 0.2 ~ 0.99 |
| Variable infiltration curve parameter | 0.001 ~ 1 |
| River width (m) | 0 ~ 101.16 |
| River depth (m) | 0 ~ 6.46 |
| River density (km/km$^2$) | 0.049 ~ 1.03 |
| River roughness | 0.033 ~ 0.05 |
| River slope | 0.015 ~ 0.47 |

*3) Line 189: How to perform the bias correction of the TIGGE-ECMWF forecast forcing*

**Response:** Thanks for your comments. We have clarified as follows:

"The resolution of TIGGE-ECMWF grid data is 0.25 °, so the data was interpolated to 5km grid to drive the CSSPv2 model. We calculated both observations' and TIGGE-ECMWF's yearly average precipitation and temperature then performed a bias correction by adding back the difference (for temperature) or multiplying back the ratio (for temperature) to match the observations' averages." (L314-L317)

*4) Line 190: Please give a detailed description of the workflow in Fig. 2.*

**Response:** Thanks for your comments. We have revised the manuscript as follows:

"Figure 2 shows the procedure of hindcasts: the calibrated CSSPv2 model was first driven with observation dataset to generate initial hydrological conditions (soil moisture, surface water, etc.) for each forecast issue date, then CSSPv2 model was driven with forecast data (TIGGE-ECMWF or ESP) at every forecast issue date with the generated initial conditions to perform a 7-day hindcast." (L231-L236)

"The network was trained on sequences of April to September in 2013-2017, with six historical streamflow observations and one forecast interval streamflow to predict the total streamflow at each forecast time step (Figure 2)." (L242-L244)

"In addition, we also tried a LSTM streamflow forecast approach which only uses 6-hour historical streamflow data as inputs, and the experiment was termed as LSTM (Table 2). The process of LSTM is similar to Meteo-Hydro-LSTM but without the forecast interval streamflow, which is also shown in Figure 2." (L250-L254)

*5) Line 204: Please check the equation number*

**Response:** The equation number has been checked and revised.

*6) Line 205: Please clarify the input and output variables here.*

**Response:** We have clarified them as follows:

"where $q_0$, $q$, $q_{max}$ and $q_{min}$ are the normalized variable, input variable, the maximum and minimum of the sequence of the variable." (L248)

*7) Lines 215: The meaning of the F(y) in Eq. 1 is not clear. Also for ref in Eq. 2. Please clarify the meaning of the variables in each equation.*

**Response:** We have revised as follows:

"The Continuous Ranked Probability Score (CRPS) was calculated as follows:

$$CRPS = \int_{-\infty}^{\infty}[F(y) - F_o(y)]^2, \qquad (7)$$

where

$$F_o(y) = \begin{cases} 0, & y < observed\ value \\ 1, & y \geq observed\ value \end{cases} \qquad (8)$$

is a cumulative-probability step function that jumps from 0 to 1 at the point where the forecast variable y equals the observation, and $F(y)$ is a cumulative-probability distribution curve formed by the forecast ensembles." (L258-L265)

*8) Line 291: Is the diurnal cycle in Fig. 8 the climatology averaged result? If so, please give the ranges to indicate the uncertainty. It also looks a bit strange that the rainfall reaches its maximum in the early morning rather than the afternoon in summer. Please clarify.*

**Response:** The time shown in the figure is the universal time. According to Beijing time zone (GMT+8:00), the rainfall reaches its peak at the afternoon, we have modified the figure and revised the descriptions as follows:

[Figure]

**Figure 8**. Diurnal cycles of Longtan outflow (m³/s; dashed black line), Yantan inflow (m³/s; solid black line) and basin-averaged precipitation (mm/h; blue line), as well as their ranges. The time shown in this figure is universal time.

*9) Lines 298-301: Why would the combination of heavy rainfall and the decrease of upstream flow make the hydrological model performance worse? Does the model consider the reservoir?*

**Response:** In this study, we used observed upstream streamflow as inputs while the ensemble meteorological forecasts as forcings. When the upstream flow decreases and rainfall increase, the upstream control on downstream flow decreases, and the influence of interval flow resulted from the meteorology-hydrology forecasts rises. And the uncertainty from meteorological forecasts would propagate to the hydrological forecasts more obviously. This highlights the importance of both the upstream outflow and the accuracy of meteorological forecasts within the catchment.

*10) Line 328: Please add some implications of this work and as well as its deficiencies that need to be improved in the last section.*

**Response:** Thanks for your comments. We have added the implications and deficiencies as follows:

"Most cascade reservoirs yet cannot forecast streamflow beyond 6 hours, and the integrated Meteo-Hydro-LSTM approach has potential to improve the forecasts at long leads. This study mainly focused on exploring the added values of meteorology-hydrology coupled forecast and LSTM forecast in a non-closed catchment, so the forecast uncertainty from upstream outflow was ignored by using the observed outflow. In the future, the upstream outflow forecast is planned to include, but this requires the development of upstream hydrometeorological forecast facility, as well as the reservoir regulation forecast that is very challenging. The artificial intelligence (AI) techniques are expected to complement the physical model for reservoir regulation." (L413-L422)

*11) Add the units in Figs. 6 and 7.*

**Response:** Revised as suggested.

[Figure]

**Figure 6.** Evaluation of precipitation and temperature hindcasts from TIGGE-ECMWF. The red and blue lines represent the best and worst results among 51 TIGGE-ECMWF ensemble members respectively, and the green lines represent the results for the ensemble means of 51 members. Solid and dashed lines represent the results after and before bias corrections, respectively.

[Figure]

**Figure 7.** (a) Continuous Ranked Probability Score (CRPS) and (b) Root Mean Squared Error (RMSE) for daily streamflow ensemble forecasts at Yantan gauge. (c) and (d) are the skill score in terms of CRPS and RMSE for Meteo+Hydro, where ESP+Hydro is used as reference forecast.

---

## Author Response (AR2)

**Response to the comments from Reviewer #4**

We are grateful to the reviewer for the constructive and careful review. We have incorporated the comments to the extent possible. The reviewer's comments are italicized and our responses immediately follow.

*The authors have enormously improved the manuscript. Thanks for the efforts. I think it can be published with a few more minor revisions.*

**Response:** We would like to thank the reviewer for the positive comments. Please see our responses below.

*1) Line124: "..aim to improving.." change to "..aim to improve.."?*

**Response:** Revised as suggested (Line 122 in the tracked version of the revised manuscript)

*2) Line150: Why use "long" as the abbreviation for "surface downward heat radiation"? You may consider deleting it as it does not appear in the following parts.*

**Response:** Revised as suggested (Line 147).

*3) Lines 156-158: "..which would generate distributed model parameters that are different within the catchment to better represent the heterogeneity of the rainfall-runoff processes". Please rewrite this sentence.*

**Response:** Thanks for your comment. We have revised as follows:

"..The calibrated runoff parameters can better represent the heterogeneity of the rainfall-runoff processes and improve runoff simulations."(Line 155-157)

*4) Line 249: Add "respectively"*

**Response:** Revised as suggested. (Line 246)

*5) Lines 283-290: "For the downstream areas with less precipitation, the higher NSE is related to the higher percentage of sand in the soil (not shown)." I doubt whether the higher NSE downstream can be explained by the spatial distribution of the sand proportion. In Figure R2, the downstream sand proportion shows a large spatial variability which is not consistent with the spatial pattern of NSE in Figure 3. It also confuses me that the higher sand content is related to better performance of the hydrological model, which seems to be contrary to the fact that the model performs better in wetter areas where the sand content is not very large. Please clarify.*

**Response:** Thanks for the comments. The higher NSE in the downstream areas can be explained by two portions. In the Jiazhuan region (the southwest region of the basin), the performance is good because of more precipitation and not very large sand contents, which is consistent to the provided fact (Figures R1, R2). The south and southeast regions have similar precipitation as the northern region. With more sand portion, rainfall in such region will infiltrate easier, and yields less runoff under infiltration excess, which is more suitable for the saturation excess-based runoff scheme in the CSSPv2 model, and shows a better performance against the northern grids.

[Figure]

**Figure R1**. The spatial distribution of average yearly precipitation (mm).

[Figure]

**Figure R2**. Spatial distribution of sand proportion.

6) *Line 317: "..the ratio (for temperature).." change to "..the ratio (for precipitation)..."?*

**Response:** Revised as suggested (Line 312).

7) *Figure 8: The title is not consistent with that in the response to comment #8.*

**Response:** Thanks for your comments. We have updated the revision (Line 632-634).